# Co-Generation with GANs using AIS based HMC

**Tiantian Fang**
University of Illinois at Urbana-Champaign
tf6@illinois.edu

**Alexander G. Schwing**
University of Illinois at Urbana-Champaign
aschwing@illinois.edu

## Abstract

Inferring the most likely configuration for a subset of variables of a joint distribution given the remaining ones – which we refer to as co-generation – is an important challenge that is computationally demanding for all but the simplest settings. This task has received a considerable amount of attention, particularly for classical ways of modeling distributions like structured prediction. In contrast, almost nothing is known about this task when considering recently proposed techniques for modeling high-dimensional distributions, particularly generative adversarial nets (GANs). Therefore, in this paper, we study the occurring challenges for co-generation with GANs. To address those challenges we develop an annealed importance sampling based Hamiltonian Monte Carlo co-generation algorithm. The presented approach significantly outperforms classical gradient based methods on a synthetic and on the CelebA and LSUN datasets. The code is available at https://github.com/AilsaF/cogen_by_ais.

## 1  Introduction

Finding a likely configuration for part of the variables of a joint distribution given the remaining ones is a computationally challenging problem with many applications in machine learning, computer vision and natural language processing.

Classical structured prediction approaches [39, 72, 75] which explicitly capture correlations over an output space of multiple discrete random variables permit to formulate an energy function restricted to the unobserved variables when conditioned on partly observed data. However, in many cases, it remains computationally demanding to find the most likely configuration or to sample from the energy restricted to the unobserved variables [66, 78].

Alternatively, to model a joint probability distribution which implicitly captures the correlations, generative adversarial nets (GANs) [25] and variational auto-encoders (VAEs) [36] evolved as compelling tools which exploit the underlying manifold assumption: a latent 'perturbation' is drawn from a simple distribution which is subsequently transformed via a deep net (generator/encoder) to the output space. Those methods have been used for a plethora of tasks, *e.g.*, for domain transfer [3, 10], inpainting [61, 83], image-to-image translation [32, 42, 31, 51, 63, 84, 87, 88], machine translation [11] and health care [67].

While GANs and VAEs permit easy sampling from the entire output space domain, it also remains an open question of how to sample from part of the domain given the remainder? We refer to this task as co-generation subsequently.

Co-generation has been addressed in numerous works. For instance, for image-to-image translation [32, 42, 31, 51, 63, 84, 87, 88], mappings between domains are learned directly via an encoder-decoder structure. While such a formulation is convenient if we have two clearly separate domains, this mechanism isn't scaleable if the number of output space partitionings grows, *e.g.*, for image inpainting where missing regions are only specified at test time.

To enable co-generation for a domain unknown at training time, for GANs, optimization based algorithms have been proposed [83, 50]. Intuitively, they aim at finding that latent sample that accurately matches the observed part. Dinh et al. [16] maximize the log-likelihood of the missing part given the observed one. However, we find that successful training of a GAN leads to an increasingly ragged energy landscape, making the search for an appropriate latent variable via back-propagation through the generator harder and harder until it eventually fails.

To deal with this ragged energy landscape for co-generation, we develop an annealed importance sampling (AIS) [58] based Hamiltonian Monte Carlo (HMC) algorithm [19, 59]. The proposed approach leverages the benefits of AIS, *i.e.*, gradually annealing a complex probability distribution, and HMC, *i.e.*, avoiding a localized random walk.

We evaluate the proposed approach on synthetic data and imaging data (CelebA and LSUN), showing compelling results via MSE and MSSIM metrics.

## 2 Related Work

In the following, we briefly discuss generative adversarial nets before providing background on co-generation with adversarial nets.

**Generative adversarial nets** (GANs) [24] have originally been proposed as a non-cooperative two-player game, pitting a generator against a discriminator. The discriminator is tasked to tell apart real data from samples produced by the generator, while the generator is asked to make differentiation for the discriminator as hard as possible. For a dataset of samples $x \in \mathcal{X}$ and random perturbations $z$ drawn from a simple distribution, this intuitive formulation results in the saddle-point objective

$$\max_{\theta} \min_{w} -\mathbb{E}_x[\ln D_w(x)] - \mathbb{E}_z[\ln(1 - D_w(G_\theta(z)))],$$

where $G_\theta$ denotes the generator parameterized by $\theta$ and $D_w$ refers to the discriminator parameterized by $w$. The discriminator assesses the probability of its input argument being real data. We let $\mathcal{X}$ denote the output space. Subsequently, we refer to this formulation as the 'Vanilla GAN,' and note that its loss is related to the Jensen-Shannon divergence. Many other divergences and distances have been proposed recently [4, 44, 26, 38, 14, 12, 56, 6, 55, 49, 28, 64] to improve the stability of the saddle-point objective optimization during training and to address mode-collapse, some theoretically founded and others empirically motivated. It is beyond the scope to review all those variants.

**Co-generation**, is *the task of obtaining a sample for a subset of the output space domain, given as input the remainder of the output space domain*. This task is useful for applications like image inpainting [61, 83] or image-to-image translation [32, 42, 31, 51, 63, 84, 87, 88]. Many formulations for co-generation have been considered in the past. However, few meet the criteria that *any* given a subset of the output space could be provided to generate the remainder.

Conditional GANs [54] have been used to generate output space objects based on a given input signal [80]. The output space object is typically generated as a whole and, to the best of our knowledge, no decomposition into multiple subsets is considered.

Co-generation is related to multi-modal Boltzmann machines [70, 60], which learn a shared representation for video and audio [60] or image and text [70]. Restricted Boltzmann Machine based encoder-decoder architectures are used to reconstruct either video/audio or image/text given one of the representations. Co-generation is also related to deep net based joint embedding space learning [37]. Specifically, a joint embedding of images and text into a single vector space is demonstrated using deep net encoders. After performing vector operations in the embedding space, a new sentence can be constructed using a decoder. Co-generation is also related to cross-domain image generation [85, 62, 18, 3]. Those techniques use an encoder-decoder style deep net to transform rotation of faces, to learn the transfer of style properties like rotation and translation to other objects, or to encode class, view and transformation parameters into images.

Image-to-image translation is related in that a transformation between two domains is learned either via an Image Transformation Net or an Encoder-Decoder architecture. Early works in this direction tackled supervised image-to-image translation [33, 32, 41, 9, 48, 10] followed by unsupervised variants [71, 68, 84, 63, 8, 81, 73, 29]. Cycle-consistency was discovered as a convenient regularization mechanism in [35, 87, 51, 2] and a distance preserving regularization was shown by Benaim and Wolf [5]. Disentangling of image representations was investigated recently [31, 42] and ambiguity in

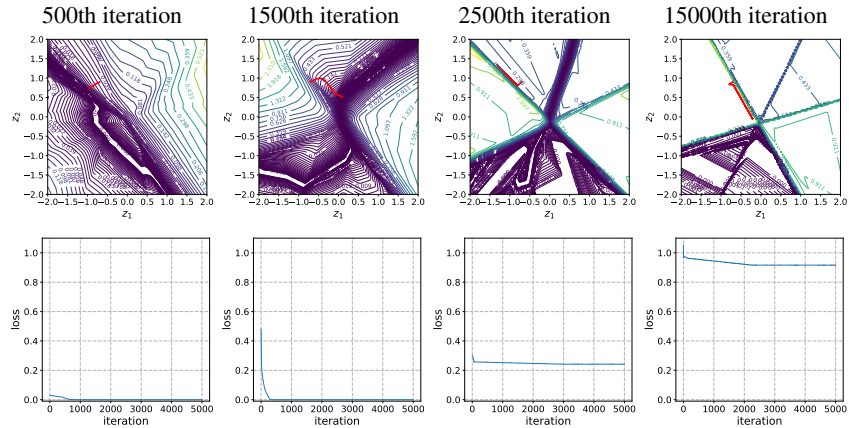

Figure 1: Vanilla GAN loss in $\mathcal{Z}$ space (top) and gradient descent (GD) reconstruction error for 500, 1.5k, 2.5k and 15k generator training epochs.

the task was considered by Zhu et al. [88]. Other losses such as a 'triangle formulation' have been investigated in [21, 43].

Attribute transfer [40], analogy learning [27, 62] and many style transfer approaches [74, 7, 79, 13, 52, 77, 30, 45, 47, 76, 17, 65, 20, 23, 86] just like feature learning via inpainting [61] are also using an encoder-decoder formulation, which maps entire samples from one domain to entire samples in another domain.

Co-generation is at least challenging if not impossible for all the aforementioned works since decoders need to be trained for every subset of the output space domain. This is not scalable unless we know ahead of time the few distinct subsets of interest.

Hence, to generate arbitrary sub-spaces, other techniques need to be considered. Some applicable exceptions from the encoder-decoder style training are work on style transfer by Gatys et al. [22], work on image inpainting by Yeh et al. [83], and coupled generative adversarial nets (CoGANs) by Liu and Tuzel [50]. In all three formulations, a loss is optimized to match observations to parts of the generated data by iteratively computing gradient updates for a latent space sample. In particular, Liu and Tuzel [50] learn a joint distribution over multiple domains by coupling multiple generators and possibly discriminators via weight-sharing. Liu and Tuzel [50] briefly discuss co-generation when talking about "cross-domain image transformation," report to observe coverage issues and state that they leave a detailed study to "future work." Instead of an optimization based procedure, we propose to use an annealed importance sampling based Hamiltonian Monte Carlo approach. We briefly review both techniques subsequently.

**Annealed importance sampling (AIS)** [58] is an algorithm typically used to estimate (ratios of) the partition function [82, 69]. Specifically, it gradually approaches the partition function for a distribution of interest by successively refining samples which were initially obtained from a 'simple' distribution, *e.g.*, a multivariate Gaussian. Here, we are not interested in the partition function itself, but rather in the ability of AIS to accurately draw samples from complex distributions, which makes AIS a great tool for co-generation.

**Hamiltonian Monte Carlo (HMC)** [19] originally referred to as "Hybrid Monte Carlo" united the Markov Chain Monte Carlo [53] technique with molecular dynamics approaches [1]. Early on they were used for neural net models [57] and a seminal review by Neal [59] provides a detailed account. In short, a Hamiltonian combines the potential energy, *i.e.*, the log probability that we are interested in sampling from with auxiliary kinetic energy. The latter typically follows a Gaussian distribution. HMC alternates updates for the kinetic energy with Metropolis updates computed by following a trajectory of constant value along the Hamiltonian to compute a new proposal. HMC is useful for co-generation because of its reduced random-walk behavior as we will explain next.

## 3 AIS based HMC for Co-Generation

In the following we first motivate the problem of co-generation before we present an overview of our proposed approach and discuss the details of the employed Hamiltonian Monte Carlo method.

## 3.1 Motivation

Assume we are given a well trained generator $\hat{x} = G_\theta(z)$, parameterized by $\theta$, which is able to produce samples $\hat{x}$ from an implicitly modeled distribution $p_G(x|z)$ via a transformation of embeddings $z$ [25, 4, 46, 14, 15]. Further assume we are given partially observed data $x_o$ while the remaining part $x_h$ of the data $x = (x_o, x_h)$ is latent, *i.e.*, hidden. Note that during training of the generator parameters $\theta$ we don't assume information about which part of the data is missing to be available.

To reconstruct the latent parts of the data $x_h$ from available observations $x_o$, a program is often formulated as follows:

$$z^* = \arg\min_z \|x_o - G_\theta(z)_o\|_2^2, \tag{1}$$

where $G_\theta(z)_o$ denotes the restriction of the generated sample $G_\theta(z)$ to the observed part. We focus on the $\ell_2$ loss here but note that any other function measuring the fit of $x_o$ and $G_\theta(z)_o$ is equally applicable. Upon solving the program given in Eq. (1), we easily obtain an estimate for the missing data $\hat{x}_h = G(z^*)_h$.

Although the program given in Eq. (1) seems rather straightforward, it turns out to be really hard to solve, particularly if the generator $G_\theta(z)$ is very well trained. To see this, consider as an example a generator operating on a 2-dimensional latent space $z = (z_1, z_2)$ and 2-dimensional data $x = (x_1, x_2)$ drawn from a mixture of five equally weighted Gaussians with a variance of $0.02$, the means of which are spaced equally on the unit circle. For this example we use $h = 1$ and let $x_o = x_2 = 0$. In the first row of Fig. 1 we illustrate the loss surface of the objective given in Eq. (1) obtained when using a generator $G_\theta(z)$ trained on the original 2-dimensional data for 500, 1.5k, 2.5k, and 15k iterations (columns in Fig. 1).

Even in this simple 2-dimensional setting, we observe the latent space to become increasingly ragged, exhibiting folds that clearly separate different data regimes. First or second order optimization techniques cannot cope easily with such a loss landscape and likely get trapped in local optima. To illustrate this we highlight in Fig. 1 (first row) the trajectory of a sample $z$ optimized via gradient descent (GD) using red color and provide the corresponding loss over the number of GD updates for the objective given in Eq. (1) in Fig. 1 (second row). We observe optimization to get stuck in a local optimum as the loss fails to decrease to zero once the generator better captures the data.

To prevent those local-optima issues for co-generation, we propose an annealed importance-sampling (AIS) based Hamiltonian Monte Carlo (HMC) method in the following.

## 3.2 Overview

In order to reconstruct the hidden portion $x_h$ of the data $x = (x_o, x_h)$ we are interested in drawing samples $\hat{z}$ such that $\hat{x}_o = G_\theta(\hat{z})_o$ has a high probability under $\log p(z|x_o) \propto -\|x_o - G_\theta(z)_o\|_2^2$. Note that the proposed approach is not restricted to this log-quadratic posterior $p(z|x_o)$ just like the objective in Eq. (1) is not restricted to the $\ell_2$ norm.

To obtain samples $\hat{z}$ following the posterior distribution $p(z|x_o)$, the sampling-importance-resampling framework provides a mechanism which only requires access to samples and doesn't need computation of a normalization constant. Specifically, for sampling-importance-resampling, we first draw latent points $z \sim p(z)$ from a simple prior distribution $p(z)$, *e.g.*, a Gaussian. We then compute weights according to $p(z|x_o)$ in a second step and finally resample in a third step from the originally drawn set according to the computed weights.

However, sampling-importance-resampling is particularly challenging in even modestly high-dimensional settings since many samples are required to adequately cover the space to a reasonable degree. As expected, empirically, we found this procedure to not work very well. To address this concern, here, we propose an annealed importance sampling (AIS) based Hamiltonian Monte Carlo (HMC) procedure. Just like sampling-importance-resampling, the proposed approach only requires access to samples and no normalization constant needs to be computed.

More specifically, we use annealed importance sampling to gradually approach the complex and often high-dimensional posterior distribution $p(z|x_o)$ by simulating a Markov Chain starting from the prior distribution $p(z) = \mathcal{N}(z|0, I)$, a standard normal distribution with zero mean and unit variance. With the increasing number of updates, we gradually approach the true posterior. Formally, we define an annealing schedule for the parameter $\beta_t$ from $\beta_0 = 0$ to $\beta_T = 1$. At every time step $t \in \{1, \ldots, T\}$ we refine the samples drawn at the previous timestep $t - 1$ so as to represent

**Algorithm 1** AIS based HMC

---

1: **Input:** $p(z|x_o)$, $\beta_t \; \forall t \in \{1, \dots, T\}$
2: Draw set of samples $z \in \mathcal{Z}$ from prior distribution $p(z)$
3: **for** $t = 1, \dots, T$ **do**                                              // AIS loop
4:     Define $\hat{p}_t(z|x_o) = p(z|x_o)^{\beta_t} p(z)^{1-\beta_t}$
5:     **for** $m = 1, \dots, M$ **do**                                          // HMC loop
6:         $\forall z \in \mathcal{Z}$ initialize Hamiltonian and momentum variables $v \sim \mathcal{N}(0, I)$
7:         $\forall z \in \mathcal{Z}$ compute new proposal sample using leapfrog integration on Hamiltonian
8:         $\forall z \in \mathcal{Z}$ use Metropolis Hastings to check whether to accept the proposal and update $\mathcal{Z}$
9:     **end for**
10: **end for**
11: **Return:** $\mathcal{Z}$

---

the distribution $\hat{p}_t(z|x_o) = p(z|x_o)^{\beta_t} p(z)^{1-\beta_t}$. Intuitively and following the spirit of annealed importance sampling, it is easier to gradually approach sampling from $p(z|x_o) = \hat{p}_T(z|x_o)$ by successively refining the samples. Note the notational difference between the posterior of interest $p(z|x_o)$, and the annealed posterior $\hat{p}_t(z|x_o)$.

To successively refine the samples we use Hamilton Monte Carlo (HMC) sampling because a proposed update can be far from the current sample while still having a high acceptance probability. Specifically, HMC enables to bypass to some extent slow exploration of the space when using classical Metropolis updates based on a random walk proposal distribution.

Combining both AIS and HMC, the developed approach summarized in Alg. 1 iteratively proceeds as follows after having drawn initial samples from $p(z) = \hat{p}_0(z|x_o)$: (1) define the desired proposal distribution; and (2) for $K$ iterations compute new proposals using leapfrog integration and check whether to replace the previous sample with the new proposal. Subsequently, we discuss how to compute proposals and how to check acceptance.

### 3.3   Hamilton Monte Carlo

Hamilton Monte Carlo (HMC) explores the latent space much more quickly than a classical random walk algorithm. Moreover, HMC methods are particularly suitable for co-generation because they are capable of traversing folds in an energy landscape. To this end, HMC methods trade potential energy $U_t(z) = -\log \hat{p}_t(z|x_o)$ with kinetic energy $K_t(v)$. Hereby the dimension $d$ of the momentum variable $v \in \mathbb{R}^d$ is identical to that of the latent samples $z \in \mathbb{R}^d$. For readability, we drop the dependence on the time index $t$ from here on.

Specifically, HMC defines a Hamiltonian $H(z, v) = U(z) + K(v)$ or conversely a joint probability distribution $\log p(z, v) \propto -H(z, v)$ and proceeds by iterating three steps $M$ times.

In a first step, the Hamiltonian is initialized by randomly sampling the momentum variable $v$, typically using a standard Gaussian. Note that this step leaves the joint distribution $p(z, v)$ corresponding to the Hamiltonian invariant as the momentum $v$ is independent of samples $z$ and as we sample from the correct pre-defined distribution for the momentum variables.

In a second step, we compute proposals $(z^*, v^*)$ via leapfrog integration to move along a hypersurface of the Hamiltonian, *i.e.*, the value of the Hamiltonian does not change. However note, in this step, kinetic energy $K(v)$ can be traded for potential energy $U(z)$ and vice versa.

In the final third step we decide whether to accept the proposal $(z^*, v^*)$ computed via leapfrog integration. Formally, we accept the proposal with probability

$$\min\{1, \exp\left(-H(z^*, v^*) + H(z, v)\right)\}. \tag{2}$$

If the proposed state $(z^*, v^*)$ is rejected, the $m + 1$-th iteration reuses $z$, otherwise $z$ is replaced with $z^*$ in the $m + 1$-th iteration.

Note that points $(z, v)$ with different probability density are only obtained during the first step, *i.e.*, sampling of the moment variables $v$. Importantly, resampling of $v$ can change the probability density by a large amount. As evident from Eq. (2), a low value for the Hamiltonian obtained after resampling $v$ increases the chances of accepting this proposal, *i.e.*, we gradually increase the number of samples with a low value for the Hamiltonian, conversely a high probability.

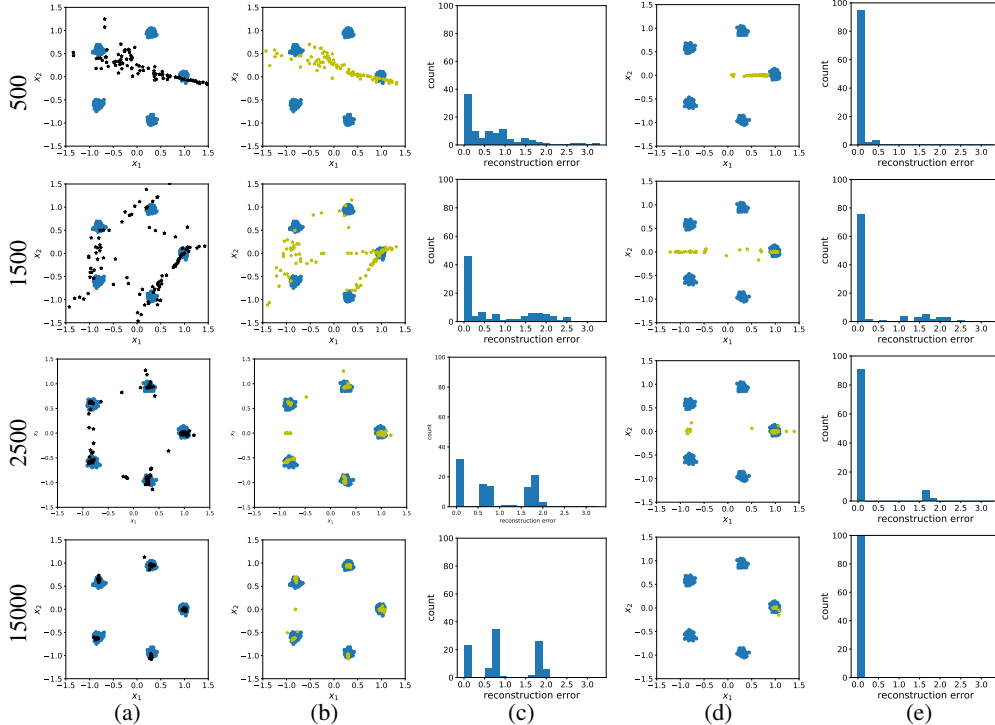

Figure 2: Rows correspond to generators trained for a different number of epochs as indicated (left). The columns illustrate: (a) Samples generated with a vanilla GAN (black); (b) GD reconstructions from 100 random initializations; (c) Reconstruction error bar plot for the result in column (b); (d) Reconstructions recovered with Alg. 1; (e) Reconstruction error bar plot for the results in column (d).

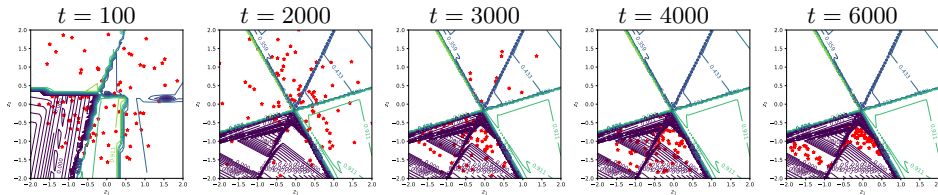

Figure 3: Samples $z$ in $\mathcal{Z}$ space during the AIS procedure: after 100, 2k, 3k, 4k and 6k AIS loops.

### 3.4 Implementation Details

We use a sigmoid schedule for the parameter $\beta_t$, *i.e.*, we linearly space $T - 1$ temperature values within a range and apply a sigmoid function to these values to obtain $\beta_t$. This schedule, emphasizes locations where the distribution changes drastically. We use $0.01$ as the leapfrog step size and employ $10$ leapfrog updates per HMC loop for the synthetic 2D dataset and $20$ leapfrog updates for the real dataset at first. The acceptance rate is $0.65$, as recommended by Neal [59]. Low acceptance rate means the leapfrog step size is too large in which case the step size will be decreased by $0.98$ automatically. In contrast, a high acceptance rate will increase the step size by $1.02$[1].

## 4 Experiments

**Baselines:** In the following, we evaluate the proposed approach on synthetic and imaging data. We compare Alg. 1 with two GD baselines by employing two different initialization strategies. The first one is sampling a single $z$ randomly. The second picks that one sample $z$ from 5000 initial points which best matches the objective given in Eq. (1).

### 4.1 Synthetic Data

To illustrate the advantage of our proposed method over the common baseline, we first demonstrate our results on 2-dimensional synthetic data. Specifically, the 2-dimensional data $x = (x_1, x_2)$ is drawn from a mixture of five equally weighted Gaussians each with a variance of $0.02$, the means of

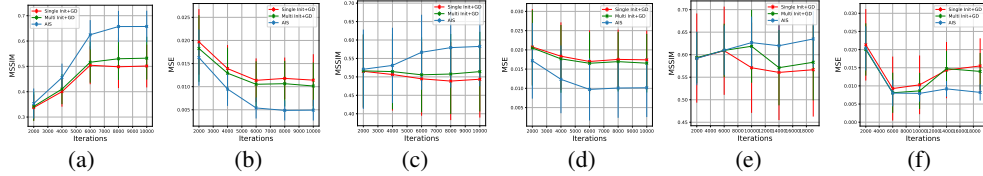

| (a) | (b) | (c) | (d) | (e) | (f) |

Figure 4: Reconstructions errors over the number of progressive GAN training iterations. (a) MSSIM on CelebA; (b) MSE on CelebA; (c) MSSIM on LSUN; (d) MSE on LSUN; (e) MSSIM on CelebA-HQ; (f) MSE on CelebA-HQ.

| Ground Truth | Masked Image | GD+single $z$ | GD+multi $z$ | AIS |

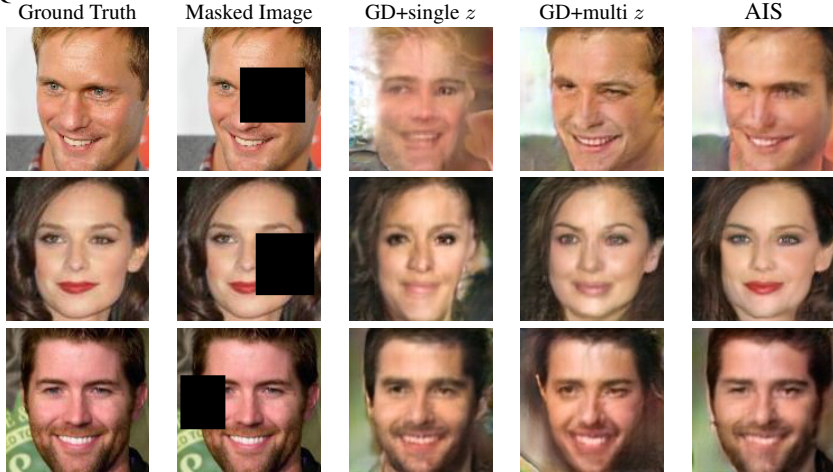

Figure 5: Reconstructions on $128 \times 128$ CelebA images for a progressive GAN trained for 10k iterations.

which are spaced equally on the unit circle. See the blue points in columns (a), (b), and (d) of Fig. 2 for an illustration.

In this experiment we aim to reconstruct $x = (x_1, x_2)$, given $x_o = x_2 = 0$. Considering the generator has learned the synthetic data very well, the optimal solution for the reconstruction is $\hat{x} = (1, 0)$, where the reconstruction error should be 0. However, as discussed in reference to Fig. 1 earlier, we observe that energy barriers in the $\mathcal{Z}$-space complicate optimization. Specifically, if we initialize optimization with a sample far from the optimum, it is hard to recover. While the strategy to pick the best initializer from a set of $5,000$ points works reasonably well in the low-dimensional setting, it obviously breaks down quickly if the latent space dimension increases even moderately.

In contrast, our proposed AIS co-generation method only requires one initialization to achieve the desired result after $6,000$ AIS loops, as shown in Fig. 2 (15000 (d)). Specifically, reconstruction with generators trained for a different number of epochs (500, 1.5k, 2.5k and 15k) are shown in the rows. The samples obtained from the generator for the data (blue points in column (a)) are illustrated in column (a) using black color. Using the respective generator to solve the program given in Eq. (1) via GD yields results highlighted with yellow color in column (b). The empirical reconstruction error frequency for this baseline is given in column (c). The results and the reconstruction error frequency obtained with Alg. 1 are shown in columns (d, e). We observe significantly better results and robustness to initialization.

In Fig. 3 we show for 100 samples that Alg. 1 moves them across the energy barriers during the annealing procedure, illustrating the benefits of AIS based HMC over GD.

## 4.2 Imaging Data

To validate our method on real data, we evaluate on three datasets, using MSE and MSSIM metrics. For all three experiments, we use the progressive GAN architecture [34] and evaluate baselines and AIS on progressive GAN training data.

**CelebA:** For CelebA, the size of the input and the output are 512 and $128 \times 128$ respectively. We generate corrupted images by randomly masking blocks of width and height ranging from 30 to 60. Then we use Alg. 1 for reconstruction with 500 HMC loops.

In Fig. 4 (a,b), we observe that Alg. 1 outperforms both baselines for all GAN training iterations on both MSSIM and MSE metrics. The difference increases for better trained generators. In Fig. 5, we

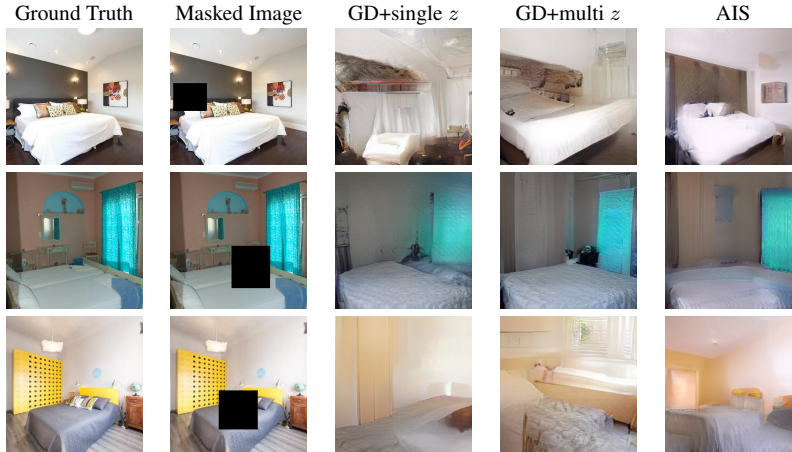

Figure 6: Reconstructions on $256 \times 256$ LSUN images using a pre-trained progressive GAN trained for 10k iter.

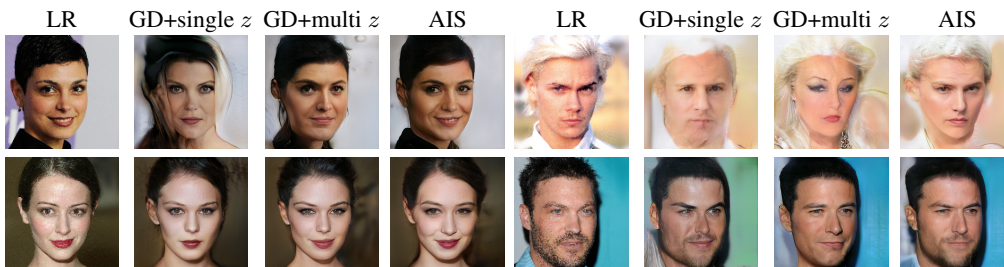

Figure 7: SISR: $128 \times 128$ to $1024 \times 1024$ for CelebA-HQ images using a progressive GAN (19k iter.).

show some results generated by both baselines and Alg. 1. Compared to baselines, Alg. 1 results are more similar to the ground truth and more robust to different mask locations. Note that Alg. 1 only uses one initialization, which demonstrates its robustness to initialization.

**LSUN:** The output size is $256 \times 256$. We mask images with blocks of width and height between 50 to 80. The complex distribution and intricate details of LSUN challenge the reconstruction. Here, we sample 5 initializations in our Alg. 1 (line 2). We use 500 HMC loops for each initialization independently. For each image, we pick the best score among five and show the average in Fig. 4 (c,d). We observe that Alg. 1 with 5 initializations easily outperforms GD with 5,000 initializations. We also show reconstructions in Fig. 6.

**CelebA-HQ** Besides recovering masked images, we also demo co-generation on single image super-resolution (SISR). In this task, the ground truth is a high-resolution image $x$ ($1024 \times 1024$) and the exposure information $x_e$ is a low-resolution image ($128 \times 128$). Here, we use the Progressive CelebA-HQ GAN as the generator. After obtaining the generated high-resolution image, we downsample it to $128 \times 128$ via pooling and aim to reduce the squared error between it and the final result. We use 3 $z$ samples for the SISR task. We show MSSIM and MSE between the ground truth ($1024 \times 1024$) and the final output on Fig. 4 (e, f). Fig. 7 compares the outputs of baselines to those of Alg. 1.

## 5 Conclusion

We propose a co-generation approach, *i.e.*, we complete partially given input data, using annealed importance sampling (AIS) based on the Hamiltonian Monte Carlo (HMC) method. Different from the classical optimization based methods, specifically GD, which get easily trapped in local optima when solving this task, the proposed approach is much more robust. Importantly, the method can traverse large energy barriers that occur when training generative adversarial nets. Its robustness is due to AIS gradually annealing a probability distribution and HMC avoiding localized walks.

**Acknowledgments:** This work is supported in part by NSF under Grant No. 1718221 and MRI #1725729, UIUC, Samsung, 3M, Cisco Systems Inc. (Gift Award CG 1377144) and Adobe. We thank NVIDIA for providing GPUs used for this work and Cisco for access to the Arcetri cluster.

## Footnotes

[1]We adapt the AIS implementation from `https://github.com/tonywu95/eval_gen`

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
