[Supplementary Material]

# Supplementary Material: Co-Generation with GANs using AIS based HMC

**Tiantian Fang**
University of Illinois at Urbana-Champaign
tf6@illinois.edu

**Alexander G. Schwing**
University of Illinois at Urbana-Champaign
aschwing@illinois.edu

## 1 Appendix: Additional Synthetic Data Analysis

Figure 1: (a) Samples generated with a vanilla GAN (black); (b) GD reconstructions from 100 random initializations; (c) Reconstruction error bar plot for the result in column (b); (d) Reconstructions recovered with Alg. 1; (e) Reconstruction error bar plot for the results in column (d).

Figure 2: WGAN-GP's loss in $\mathcal{Z}$ space and GD loss for reconstruction at 1000th, 5000th, 10000th and 50000th epoch.

Sampling from a multi-modal distribution is challenging. Particularly if the modes are well separated it is important to adequately explore the domain in order not to get stuck in a single mode. To observe this we study the ability to sample from a multi-modal distribution on our synthetic data. We use observation $x_o = x_1 = -1$ which retains an ambiguous $x_2 = 0.5$ or $x_2 = -0.5$. For this experiment, we use the same 15k-th iteration generative model as employed in Fig. **??**. Results for GD are shown

Figure 3: Rows correspond to generators trained for a different number of epochs as indicated (left). The columns illustrate: (a) Samples generated with a WGAN-GP (black); (b) GD reconstructions from 100 random initializations; (c) Reconstruction error bar plot for the result in column (b); (d) Reconstructions recovered with our proposed AIS based HMC algorithm; (e) Reconstruction error bar plot for the results in column (d).

Figure 4: $z$ state in $\mathcal{Z}$ space during the AIS procedure after the 100th, 3700th, 3800th, 3900th and 4000th AIS loop.

in Fig. 1 (b,c) while AIS results are provided in Fig. 1 (d, e). We observe GD gets stuck in bad modes. In contrast, AIS finds the optimal modes.

Following Fig. 1, Fig. 2 and Fig. 3 in the main paper, we compare the performances of baselines and our proposed AIS based HMC algorithm on WGAN-GP. The results illustrate the robustness of our algorithm on a variety of GAN losses and advantages over the baselines.

The optimal solution of this experiment is $(x_1, x_2) = (1, 0)$. Fig. 2 shows energy barriers which challenge optimization w.r.t. $z$ particularly for well trained generators. Fig. 3 shows that baselines get stuck in a local optimum easily. In contrast, the reconstructions obtained by our proposed algorithm overcome the energy barriers and find good solutions within 4,000 AIS loops. We observe that the reconstruction accuracy (Fig. 3 (d, e)) increases with generator model improvements. Fig. 4 shows how 100 $z$ samples move in $\mathcal{Z}$-space and cross the barriers during the AIS procedure.

Figure 5: Reconstructions errors over the number of progressive GAN training iterations. (a) MSSIM on LSUN test data; (b) MSE on LSUN test data.

## 2 Appendix: Additional Real Data Examples

We show additional results for real data experiments. We observe our proposed algorithm to recover masked images more accurately than baselines and to generate better high-resolution images given low-resolution images.

We show masked CelebA (Fig. 6) and LSUN (Fig. 7) recovery results for baselines and our method, given a Progressive GAN generator. Note that our algorithm is pretty robust to the position of the $z$ initialization, since the generated results are consistent in Fig. 6. In Fig. 8, we compare our results with baselines on generating a $1024 \times 1024$ high-resolution image from a given $128 \times 128$ low-resolution image.

We also run both baselines and AIS on LSUN test data. The result is shown in Fig. 5. However, for both CelebA and CelebA-HQ, Progressive GANs are trained on the whole dataset, we are unable to do the experiments on these two datasets.

Figure 6: Reconstructions on $128 \times 128$ CelebA images for a trained progressive GAN at 10k-th iteration. (a) Ground truth and masked (observed) images (top to bottom); (b) The result obtained by optimizing the best $z$ picked from 5,000 initializations (top to bottom); (c) Result generated by our algorithm.

Figure 7: Reconstructions on $256 \times 256$ LSUN images for a trained progressive GAN at 10k-th iteration. (a) Ground truth; (b) Masked (observed) images; (c) The result obtained whenn optimizing a single $z$; (d) The result obtained by optimizing the best $z$ picked from 5,000 initializations; (e) Result of our algorithm.

| (a) | (b) | (c) | (d) |

Figure 8: Simple Super Resolution Task result from a $128 \times 128$ to a $1024 \times 1024$ image for a trained progressive GAN at 19k-th iteration. (a) Ground truth; (b) The result obtained by optimizing a single $z$; (c) The result obtained by optimizing the best $z$ picked from 5,000 initializations; (d) Results of our algorithm.