[Reviews · NeurIPS 2019]

Reviewer 1



# Strong points * The paper is very well written and easy to follow * The observation that inference for GANs is hampered by local minima is important and might have important consequence, e.g. regarding experiments that examine the mode coverage of GANs by inferring latent codes for test data. * The proposed solution is quite easy to implement as it is based on standard HMC * The authors attach code in the supplementary material # Weak points * It was not clear to me if HMC is really necessary or if noisy gradient descent where the noise is gradually annealed would work just as well. It would be nice to have this as a baseline (this baselines is even easier to implement than the method proposed in this paper). * While inpainting and super resolution are certainly interesting tasks, I'm not sure if these are really the most compelling examples, as it is relatively easy to generate data for a fully supervised method in these cases (e.g. for inpainting just remove random rectangles). Other interesting examples might include ill-posed problems like inverting a Radon transform for medical data or even doing multiple things at once with one model. * It would be nice to have a better ablation study / study of hyperparameters (annealing schedule, acceptance rate, leapfrog updates, step size). * It would be nice to have FID [1] as a second, more global metric, besides MSE and SSIM. * I think AIS is misnomer in this context, since there is no "importance sampling" involved (A better name might just be "annealed HMC"). # Other comments / question * Fig. 4 is very small and it would be good to replace it with a table. It would also be nice, if Fig.4 was referenced much earlier in Sec. 4.2 * "The complex distribution challenges exploration." (l. 261-262): How is the complexity of the distribution related to exploration issues. It would be nice if the authors could elaborate on this point. Are there simple experiments that show this? * "We show MSSIM and MSE between the ground truth and the final output" (l. 272-273): is this just one image or a full test set? How as the train / val / test split done? (for all datasets) * Why is gradient descent in latent space always called "SGD" and not just "GD"? What is stochastic about it? * It would be nice to have error bars for Fig. 4, i.e. standard deviation over the different test images # Overall rating While the idea in this paper is relatively simple, the overall observation that gradient descent in latent space for GANs might lead to local minima is interesting and the proposed solution is elegant and easy to implement. While I do not recommend acceptance at this stage, I would be willing to upgrade my rating if the authors can successfully address my remaining concerns and the other reviewers argue for acceptance. [1] Heusel, Martin, et al. "Gans trained by a two time-scale update rule converge to a local nash equilibrium." Advances in Neural Information Processing Systems. 2017. === UPDATE AFTER REBUTTAL === I think the authors addressed most of my original concerns (simple baseline, ablations study). Although they did not address my concern about alternative metrics, the naming of the method (I still don't see why there should be "importance sampling" in the name) and additional tasks, I don't think these concerns are major blockers. However, I think not using a test set for these kinds of experiments is very misleading. Given that the other reviewers were very positive and the paper is mostly about the optimization method, I will increase my overall score to "6 - Marginally above the acceptance threshold" . However, I strongly encourage the authors to rerun their experiments with a separate test set or at least clearly state in the paper that they did not use a test set.

Reviewer 2



Originality: the method proposes an application of annealed importance sampling and HMC to the problem of co-generation, relying on a pretrained GAN. As far as I’m aware, it is both a novel application of these algorithms, and a problem that has been seldom addressed relying on such generative models (solely pretrained for image generation, not explicitly for special cases of co-generation). Quality: + The analysis and experiments on the toy dataset highlight the issue that is addressed and provide intuitions of the pitfalls of SGD and of what the method is accomplishing in small dimension, providing a strong motivation for the proposed approach + The experiments compare the method to SGD both qualitatively and quantitatively, on three real world datasets, both for image inpainting and super-resolution tasks. These experiments show promising results, where the robustness of their proposed algorithm is highlighted in comparison with the SGD baselines. Some remarks: - If I understand correctly, the second SGD baseline consists in sampling N initializations, choosing the best in terms of the loss, and then running the optimization (according to l.223-224). It would strengthen the paper to instead run N optimizations and keep the one that has reached the best result. If the phenomena observed in the toy setting occur in large dimension as well, this should in any case yield similar results to the current baseline, but would be a stronger (and very reasonable) baseline. In particular, the LSUN experiments do this for N=5 for the proposed algorithm, so I’d like to see corresponding results for SGD. l.264-265 should also be clarified if my understanding is correct, because it is misleading (unfair comparison). Additionally, this should be made clear in the caption of Figure 4. Same remarks for CelebA-HQ, where the proposed algo uses 3 initializations. - it would be informative to give a comparison in terms of running time. In particular, the second baseline (as well as the proposed improvement) can run batches of initialisations / optimisations in parallel, while the proposed method is sequential. - there are many techniques that have been (separately) developed for super-resolution and for image inpainting. The interest of the method is in providing a unified way to address co-generation, and by leverageing a generative model that has been trained solely for the task of image generation, so in this, it is quite novel wrt the soa. Nevertheless, it would be more relevant here to quite a small overview of the techniques that have been used in each of these domains, than the overview that is given for image-to-image translation techniques, which are only vaguely related. - it would be also relevant to cite Dinh et al. [5] who propose flow based models, and investigate their use for co-generation (by maximizing the log likelihood, which can be computed exactly for their model) Clarity: the paper is extremely well written and structured, it is a pleasure to read. Some remarks: - “Cogeneration is the task of obtaining a sample for a subset of the output space domain, given as input the remainder of the output space domain“. But the technique seems to be used to re-generate the whole image eg. Figures 5 and 6. I imagine that this is because it would otherwise lead to unpleasant border effects. Nevertheless, the task should be addressed as it is defined, so this should be acknowledged, and future directions should at least be provided to address this issue. - l.89-90 "Co-generation is at least challenging if not impossible for all the aforementioned works since decoders need to be trained for every subset of the output space domain" - this is not true for the approaches that perform image inpainting, such as the work of Pathak et al. that is mentioned above. While it is indeed only trained to fill in subsets in the form of a contiguous masks, these masks are sampled randomly in location and even in shape. Significance: The proposed method has two important interesting properties, that differentiate it from competitors: + it can be generically applied to any co-generation task in a unified manner, and is demonstrated on both tasks of super-resolution and image in-painting, which are usually addressed by separate lines of work. This does not require training separate models. This suggests opportunities for other applications to be explored. + it is based on a generative model solely trained for image generation task. While discriminative and generative models can be directly trained for the task of inpainting (eg [1,2]) - and obtain better results than those shown - it is nevertheless very interesting to investigate and propose methods for application of generative adversarial networks trained for image generation, for instance to investigate usefulness of the latent representation that is learned by such methods. It could be more explicitly motivated as a way to obtain an inverse mapping that projects data into the latent space learned by a GAN, and compared to approaches like BiGAN [3] and ALI [4] (which require training an inference model during the training of the generative model). [1] Contextual-based Image Inpainting: Infer, Match, and Translate, Song et al. ECCV 2018 [2] Generative Image Inpainting with Contextual Attention, Yu et al. CVPR 2018 [3] ADVERSARIAL FEATURE LEARNING, Donahue et al. ICLR 2017 [4] ADVERSARIALLY LEARNED INFERENCE, Dumoulin et al. ICLR 2017 [5] NICE: NON-LINEAR INDEPENDENT COMPONENTS ESTIMATION, Dinh et al. ICLR 2015 ====== Updated review ====== The authors have addressed most of my concerns, but like the other reviewers, I feel that while it is of interest, this paper is of medium significance; in particular because it does not compare against strong competitors of the fields of inpainting or super-resolution; and as I pointed out in my review, does not either address exactly the task of co-generation defined. I agree with R1’s criticism about the data splits, which I had not paid attention to. It does bother me as well that the experiments have not been carried out on a different set of images than those used for training the GAN; while I don't expect the results would be entirely different, it would be better to know for sure. The authors’ justification for this (“we follow Progressive GAN”) doesn’t convince me, given that these papers address entirely different problems. I agree with R1's that the authors should at the very least make this clear in the paper. For this reason, I am decreasing my score to “Marginally above the acceptance threshold.” To conclude, for the sake of the interest of the paper, I am still recommending acceptance, but I also expect that the impact of this paper might be limited by these weaknesses.

Reviewer 3



The paper shows why image inpainting (or more generally "co-generation", though the non-toy experiments in the paper are all about inpainting) is not straightforward using GANs. It then proposes a new optimization method to solve the inpainting problem and convincingly demonstrates that it outperforms a naive approach. The paper is clearly written and easy to understand, though slightly sloppy in some places (see details below). I believe the contribution is probably meaningful enough to be accepted. My main criticism is that the paper only compares against a naive baseline, but does not adequately compare against alternative methods for inpainting. I'm not an expert in this topic, but googling "GAN inpainting" yields multiple seemingly relevant papers that this paper could compare against. The writing is slightly sloppy and imprecise in places. Annealed importance sampling is misspelled on page 3. The experiments in section 4 compare against "SGD", but it looks like this is just (non-stochastic) gradient descent. EDIT: following the reviewer discussion I recommend this paper to be accepted for a poster presentation

[Author Response · NeurIPS 2019]

We thank all reviewers for useful feedback.

**To Reviewer 1:**

Figure 1: Error bar plots for GAN after 15k iterations on synthetic data (following Fig. 2 in main paper). AIS-HMC performs best.

(a) GD  (b) NGD  (c) AIS-HMC

*Re. noisy gradient descent (NGD) baseline:* Thanks for the
suggestion. We ran this baseline on synthetic (Fig. 1) and image
(Fig. 4) data, using sigmoid annealing schedule. We observe NGD to
improve over GD on synthetic data. This is intuitive as NGD is better
able to escape some local optima. However, even on synthetic data it performs nowhere close to the AIS-HMC method.
*Re. exploration of complex distributions:* Sampling from a multi-modal distribution is challenging. Particularly if the
modes are well separated it is important to adequately explore the domain in order not to get stuck in a single mode. To
observe this we study the ability to sample from a multi-modal distribution on our synthetic data. We use observation
$x_o = x_1 = -1$ which retains an ambiguous $x_2 = 0.5$ or $x_2 = -0.5$. Results are shown in Fig. 3.
*Re. other applications:* While other applications are also insightful, we think that synthetic, inpainting and super
resolution already display the major challenges: the loss is ragged when optimizing w.r.t. the latent variable. AIS-HMC
addresses this ill-posed challenge well, particularly also ambiguity (see Fig. 3 where there exists more than one correct
prediction given $x_o = x_1 = -1$).

Figure 2: Ratio of trials reaching a reconstruction error less than 0.2: (a) leap frog step size over the number leap frog iterations and (b) the number of leap frog iterations over the number of intermediate distributions in AIS.

(a)  (b)

*Re. ablation study:* We perform two studies and show results in Fig. 2: (1) leap
frog step size over leap frog iterations; and (2) leap frog iterations over number
of intermediate AIS distributions, i.e., the number of HMC iterations. From (1)
we see the method is stable. This is due to HMC adjusting leapfrog step size and
acceptance probability. From (2) we note that performance is suboptimal with
few intermediate AIS distributions. With one AIS distribution, we run vanilla
HMC. This shows that AIS-based HMC has a big advantage over just HMC.
23
*Re. FID:* FID is not an appropriate metric for co-generation. We are interested in
retrieving a reconstruction which best fits the given data, hence accuracy matters.
Note, diversity does not necessarily exist. In contrast FID assesses diversity (among others). This being said, we do
agree that research on how to better assess this task is necessary. This is however beyond the scope.
*Re. AIS and necessity:* It is AIS-based HMC because we use an annealing process to move samples from a tractable
distribution to the target distribution via a sequence of intermediate steps (line 4 in the algorithm). HMC would
directly sample from the target distribution. For complex distributions obtained from GANs we found plain HMC to be
challenging (see Fig. 2b when the number of intermediate distributions in AIS=1).
*Re. Fig. 4 (paper):* It is first referenced five lines after discussing the experiment (L254). We'll present better. Error
bars and additional baselines are shown in Fig. 4 (rebuttal).
*Re. SGD:* Thanks for pointing out, it should be GD, we'll revise.
*Re. train/test data:* We follow and use prior work, e.g., Progressive GAN. They do not split train/test data. We use 100
images for the metric evaluation for all real image tasks. We added error bars to Fig. 4 (rebuttal).

**To Reviewer 2:**

*Re. optimizing N points:* Thanks for suggesting. This improves slightly compared to gradient descent, but is computa-
tionally more expensive. The AIS-HMC method still has a significant edge. See 'MultiOpt+GD' in Fig. 4(c, d).
*Re. run-time:* For CelebA data, AIS-HMC takes approximately 13min (0.1min for one HMC step). GD and NGD both
take approximately 15min for the 30,000 GD update iterations that we use.
*Re. other techniques & clarity:* We'll discuss, add references, e.g., to Dinh et al., and clarify.

**To Reviewer 3:**

*Re. inpainting & writing:* Please note that this isn't an inpainting paper. We are interested in studying the co-generation
task, i.e., how to optimize w.r.t. latent samples. This is more general than inpainting. Agreed, specific methods can be
trained for each task, however, this isn't the point. Thanks for pointing out writing, we'll clarify.

Figure 3: Columns illustrate: (a) Samples generated with a vanilla GAN (black); (b) GD reconstructions from 100 random initializations; (c) Reconstruction error bar plot for the result in column (b); (d) Reconstructions recovered with Alg. 1; (e) Reconstruction error bar plot for the results in column (d). (f) NGD reconstructions from 100 random initializations; (g) Reconstruction error bar plot for the result in column (f).

(a) higher is better  (b) lower is better  (c) higher is better  (d) lower is better

Figure 4: Reconstruction error over Progressive GAN training iterations: (a) MSSIM on CelebA; (b) MSE on CelebA; (c) MSSIM on LSUN; (d) MSE on LSUN.

[Meta-Review · NeurIPS 2019]

This is a purely empirical study that considers a problem of co-generation in the context of deep unsupervised generative models. Given a part of the example is observed, one is required to fill in the remaining (unobserved) part in a reasonable way. The problem is well motivated by applications such as image in-painting. The authors provide an extensive overview of the existing literature. The proposed solution is simple and uses an already trained GAN generator $G: Z \to X$ to find latent vectors $z$ resulting in outputs $G(z)$ looking similar to the observed part of the image. The authors demonstrate empirically that a naive solution optimizing $z$ with a gradient descent in the latent space often gets stuck in local minima and does not work. The paper proposes to replace the gradient descent with another optimization technique based on annealed importance sampling, which is simple to implement. The resulting method is demonstrated to significantly outperform the naive baseline in the controlled toy scenario (Figs 1, 2, 3) as well as on more reasonable and challenging data sets (Figs 4, 5). The paper is very well polished, the presentation is clear and easy to follow. I would recommend acceptance, however I would ask the authors to address all of the feedback provided by the reviewers in the rebuttal (this should not be too difficult), especially the one by Revewer 1 regarding the train/test partitioning.